# Developing a New Constitutive Model of High Damping Rubber by Combining GRU and Attention Mechanism

**DOI:** 10.3390/polym16050567

**Published:** 2024-02-20

**Authors:** Feng Li, Tianbo Peng

**Affiliations:** 1College of Civil Engineering, Tongji University, Shanghai 200092, China; lifeng19@tongji.edu.cn; 2State Key Laboratory of Disaster Reduction in Civil Engineering, Tongji University, Shanghai 200092, China

**Keywords:** high damping rubber, constitutive model, time series characteristic, GRU, attention mechanism

## Abstract

High damping rubber (HDR) bearings are extensively used in seismic design for bridges due to their remarkable energy dissipation capabilities, which is critical during earthquakes. A thorough assessment of crucial factors such as temperature, rate, experienced maximum amplitude, and the Mullins effect of HDR on the mechanics-based constitutive model of HDR is lacking. To address this issue, we propose a deep learning approach that integrates the Gate Recurrent Unit (GRU) and attention mechanism to identify time series characteristics from compression-shear test data of HDR specimens. It is shown that the combination of GRU and attention mechanism enables accurate prediction of the mechanical behavior of HDR specimens. Compared to the sole use of GRU, this suggested method significantly reduces model complexity and computation time while maintaining good prediction performance. Therefore, it offers a new approach to constructing the HDR constitutive model. Finally, the HDR constitutive model was used to analyze the impact of experienced maximum amplitudes and cycles on following processes. It was observed that maximum amplitudes directly influence the stress-strain relationship of HDR during subsequent processes. Consequently, a solid foundation is laid for evaluating the responses of HDR bearings under earthquakes.

## 1. Introduction

High damping rubber (HDR) bearings are extensively used in seismic design for bridges due to their remarkable energy dissipation capabilities. The mechanical properties of HDR play a crucial role in ensuring stable seismic effectiveness during earthquakes. It is important to note that the energy dissipation performance of HDR can be influenced by various factors. The influence of temperature on the mechanical properties of HDR bearings mainly includes: (i) instantaneous variation of HDR material at ambient temperature; (ii) the crystallization hardening of HDR material caused by long-term exposure to a low-temperature environment; (iii) under cyclic loading, the internal temperature of HDR material increases due to friction effects; (iv) HDR material aging caused by seasonal temperature changes throughout its entire life [1,2,3,4]. In [5], the mechanical performance changes of a HDR bearing were studied in the temperature range of −20 °C to 40 °C, where the experiment was conducted under a vertical compressive stress of 12 MPa and a horizontal shear strain amplitude of 100%. The results indicated that as the temperature increased, the hysteresis energy dissipation, yield strength, and equivalent damping rate of HDR bearings significantly decreased. Meanwhile, a fourteen-day aging test was conducted under high-temperature aging conditions of 100 °C. The results indicated that the high-temperature aging of the bearing reduced its mechanical properties by 2–8%. In [6], simple shear, single-step, and multi-step stress relaxation tests were conducted on HDR bearings at different temperatures of −30 °C, −10 °C, and 23 °C, respectively, to investigate the low-temperature dependence of mechanical properties and temperature dependence of viscosity of HDR bearings. The simple shear test results indicated that as the temperature decreased, the hysteresis area of the rubber bearing increased. When the HDR bearings were sheared to strains of 100%, 150%, and 175%, the single step relaxation test results showed that the viscosity of the HDR bearings had a strong temperature dependence. In [7], the influence of performance changes of HDR material on the seismic response of base isolation structures was studied. The results indicated that the dissipated energy of HDR material caused temperature rises of 10 °C to 20 °C, at ambient temperatures of -10 °C and 20 °C. The results indicated that the damping force and stiffness of HDR material significantly affected the seismic response of base isolation structures with temperature changes. The maximum displacement difference of structural response can reach 10%, and the maximum inter-story displacement difference can reach 20%. In [8], uniaxial tensile tests (strain range 0–80%, strain rate 2 × 10^−1^ s^−1^, 2 × 10^−2^ s^−1^, 2 × 10^−3^ s^−1^ and 2 × 10^−4^ s^−1^) and cyclic tensile tests (strain range 20–80%, strain rate ± 2 × 10^−1^ s^−1^) were conducted on carbon black filled rubber materials at −20 °C, 23 °C, 60 °C, and 100 °C, respectively. The results indicated that as the temperature increased, the hysteresis loop area of the rubber material decreased. In [9], the influence of compressive load on HDR rate-dependent behavior was clearly emphasized by conducting simple shear tests at different strain rates and compressive stress. When the compressive stress was within the range of 8 MPa to 12 MPa, the maximum shear stress at all strain rates increased by an average of about 20%, with a more significant increase at lower strain rates. In contrast, the shear stress obtained from the multi-step relaxation test showed almost no difference, indicating that the influence of compressive load on the equilibrium response of HDR can be ignored. In [10], a new type of rubber-like material with shear resistance was introduced to study the mechanical properties and shear constitutive model of the material. The vertical bearing capacity and horizontal shear performance of isolation bearings with four different design parameters were evaluated. Further research on the mechanical properties of this type of rubber material indicated that it had stable yield stress and strong rate dependent characteristics. Based on the experimental results, a rheological shear constitutive model of the material was proposed to describe its highly nonlinear rate-dependent characteristics. In order to identify model parameters, parameter identification experiments were conducted, including multi-step relaxation tests, cyclic shear tests, and single-step relaxation tests. Finally, the accuracy of the established constitutive model was verified through bearing tests. The results indicated that the proposed rheological model accurately describes the highly nonlinear and rate-dependent shear mechanical properties of the bearing. Due to the large deformation and viscoelasticity of rubber-like elastomers, establishing their rate-dependent constitutive relationships is very challenging. In [11], the energy dissipation rate of material was derived from the basic theory of thermodynamics. Based on viscoelastic theory and the energy dissipation rate, a viscoelastic constitutive relationship suitable for describing the rate-dependent behavior of rubber-like elastomer was proposed. Meanwhile, through comparative analysis, it was found that the overall prediction accuracy of the new model was better than that of some commonly used models. Finally, the nonlinear mechanical behavior of three types of rubber elastomers under different strain rates was predicted using the newly proposed constitutive relationship. In [12], the mechanical response of HDR under different loading conditions was studied. By making four specimens with a shear strain of 300% and a displacement capacity of 75 mm, each specimen was subjected to harmonic loading, and the effects of strain amplitude, loading frequency, strain history, and cyclic loading on the mechanical properties of HDR were studied. In addition, an analytical model was proposed to simulate the HDR behavior under different strain amplitudes and loading frequencies. The experimental results showed that the developed HDR provided a damping rate of 15% to 43% under different loading conditions, and had a relatively stable hysteresis curve under cyclic loading. In addition, without increasing stiffness, the energy dissipation capacity of the damper was significantly improved when the loading frequency was greater than 0.5 Hz. The stress performance of HDR bearings is very complex. The currently available models are generally limited to unidirectional motion, and in most cases, it is difficult to extend to general bidirectional loading. One of the main limitations is that they cannot describe the behavior of bearings under different levels of shear deformation. In [13], a bidirectional model was proposed to analyze the shear performance of HDR bearings under general bidirectional loads, and calibrated using a set of unidirectional harmonic tests with strain amplitudes ranging from 5% to 200%. This model has been qualitatively validated on a series of bidirectional experimental data and used for free vibration and bidirectional seismic simulation of base isolated buildings. The implementation of the proposed model is simple and requires a unique set of parameters to characterize the response of the support under low, medium, and high shear amplitudes.

Scholars have extensively researched the HDR constitutive model, primarily focusing on two types: the molecular chain network model based on thermodynamic statistical theory [14,15] and the phenomenological theory model based on continuum mechanics [16,17,18,19,20]. However, many of these studies focus solely on developing theoretical expressions to describe the loading and unloading stages of the HDR constitutive model, without fully considering the impact of crucial factors such as temperature, rate, and maximum amplitude on the model.

In addition to the aforementioned factors, it is important to note that rubber materials display a significant stress-softening phenomenon under initial cyclic loads. This phenomenon, commonly referred to as the ‘Mullins effect’, was first discovered by Holt [21]. Since then, it has been extensively studied by Mullins, etc. [22,23,24,25]. The Mullins effect is enhanced by the addition of carbon black in HDR. Accurately simulating the influence of various primary factors on the HDR constitutive model, while considering the Mullins effect, is a challenging task in practical engineering applications. The study of the HDR constitutive model with complex nonlinear relationships has significant theoretical and practical significance.

Machine learning [26,27,28] methods do not require mechanical hypotheses, unlike methods based on mechanical theory. They utilize the computing and data processing power of computers to solve complex problems in engineering and science. However, in the era of big data, traditional machine learning faces the challenge of processing massive amounts of information, which demands higher data processing ability and computing speed from machine learning algorithms. The development of deep learning algorithms [29,30,31] has enabled the effective handling of complex data types and tasks.

Compression-shear tests are conducted on HDR specimens under different conditions to accurately describe the complex nonlinear HDR constitutive model. An HDR constitutive model is developed using the Recurrent Neural Network (RNN) [32] that incorporates the influences of temperature, rate, experienced maximum amplitude, and the Mullins effect of HDR. The test data verifies the model’s generalization ability, demonstrating the feasibility of developing the HDR constitutive model using deep learning methods. The HDR constitutive model is then utilized to investigate the impact of experienced maximum amplitudes and cycles.

## 2. Compression-Shear Test

When HDR bearings are used as seismic isolation devices, the forces acting on them are vertical compressive force and horizontal shear force. Therefore, a compression-shear test is conducted, taking into account factors such as temperature, rate, and experienced maximum amplitude. This test is performed by the code described in [33].

### 2.1. HDR Specimen and Loading Devices

The HDR specimen is composed of two steel plates measuring 250 mm × 250 mm × 20 mm, with a single HDR layer measuring 250 mm × 250 mm × 5 mm sandwiched between them. The steel plates and HDR layer are bonded together through vulcanization, as illustrated in Figure 1. For temperature control, the high and low temperature alternating environment test chamber manufactured by Shanghai DOAHO Test Equipment Factory (Shanghai, China) is used. The 1000 kN electro-hydraulic servo actuator provided by the MTS company (Eden Prairie, MN, USA) is used as the horizontal loading equipment, while the 2000 kN vertical actuator manufactured by the FCS company (Beijing, China) is used as the vertical loading equipment, as shown in Figure 2.

### 2.2. Test Cases

To examine the effects of temperature, rate, and maximum amplitude on the RNN model, two separate cases are designed for training and testing, as shown in Table 1 and Table 2. To avoid producing electrification of rubber under mechanical action, all equipment is grounded during the test process.

It has been observed that the vertical compressive force has minimal influence on the hysteretic performance of the HDR specimen. Therefore, a vertical compressive force of 625 kN is applied for all test cases, resulting in a compressive stress of 10 MPa. To examine the impact of the experienced maximum amplitude on the subsequent process, we conducted two specific tests: amplitude-increasing cyclic loading (40%→80%→120%→160%→200%) and amplitude-decreasing cyclic loading (200%→160%→120%→80%→40%). Each amplitude underwent three cycles. The stress-strain relationships are illustrated in Figure 3, with a temperature of 283.15 K and a loading rate of 4 mm·s^−1^.

From Figure 3, it is evident that the Mullins effect is eliminated for small amplitudes as the loading amplitude increases. Similarly, for large amplitudes, the Mullins effect is eliminated immediately upon decreasing cyclic loading, and subsequent loading does not exhibit the Mullins effect. The experienced maximum amplitude by the HDR specimen significantly influences its subsequent mechanical properties, which exhibit clear time series characteristics.

It is important to note that the mechanical properties of the specimen would recover after twenty-four hours, indicating the inevitable occurrence of the Mullins effect when HDR bearings are used for seismic isolation.

## 3. RNN Model Development

The constitutive model of HDR exhibits clear time series characteristics, requiring a comprehensive investigation using a modeling methodology capable of incorporating the preceding information. In its simplest form, the conventional neural network encounters challenges in considering the influence of previous information on the present state. RNN possesses the ability to retain and use previous information, which significantly influences the current state. The RNN expansion structure [34], as illustrated in Figure 4, consists of a set of neural networks denoted by A. The input and output are represented by *x_t_* and *h_t_*, respectively. However, RNN fails to address the issue of long-term dependencies [35]. This issue is effectively resolved by Long Short Term Memory networks (LSTM) [36], which is a specialized variant of RNN. The Gate Recurrent Unit (GRU), a simplified adaptation derived from LSTM, is capable of achieving comparable performance to LSTM [37].

### 3.1. GRU Model

This study utilizes GRU due to its computational advantages for learning the test data of HDR specimens. The GRU model was developed using the TensorFlow-GPU-2.6.0 deep learning framework, with the CUDA 11.2 computing platform and cuDNN 8.1 GPU acceleration library of the deep neural network to expedite operations on the NVIDIA RTX A4000 (Shanghai, China) graphics card. The overall flowchart s shown in Figure 5.

Step (1): Data Acquisition. The data collected include time, temperature, shear strain, shear strain rate, amplitude, and shear stress. The data cleaning process comprises several steps, including data smoothing, unification of case data, and data standardization. Data standardization is achieved using the Z-Score method, as shown in Equation (1).
(1)$$z=\frac{x-\mu }{\sigma },$$
where *z* is the standardized data, *x* is the original data, *μ* is the mean value of the original data, and *σ* is the standard deviation of the original data.

After completing the aforementioned data processing steps, it is essential to segment the data into sequential data for GRU input using a sliding window technique with a length of fifty data points.

Step (2): Model Training. During this process, time, temperature, shear strain, shear strain rate, and amplitude are taken as inputs, and shear stress as output. The GRU parameters, optimizer, and early stopping epochs are shown in Table 3. The loss function used in this study is Mean Square Error (MSE), calculated according to Equation (2).
(2)$$MSE=\frac{1}{n}\sum_{i=1}^{n}{\left(Y_{i}-{\hat{Y}}_{i}\right)}^{2},$$
where *n* is the number of samples, $Y_{i}$ is the test value, and ${\hat{Y}}_{i}$ is the predicted value.

Step (3): Model Testing. During model testing, the batch size used is the same as the batch size employed during model training. The evaluation metric selected for assessing predictive performance is the determination coefficient *R*^2^, which is calculated using Equation (3).
(3)$$R^{2}=1-\frac{MSE(Y,\hat{Y})}{Var(Y)},$$
where Y is the test value, $\hat{Y}$ is the predicted value, and Var(Y) is the variance.

By repeating steps (2) and (3), the model can undergo further training and testing to continuously improve, until achieving the desired prediction results.

Step (4): Model Tuning. To address the issue of over-fitting, we progressively changed the GRU model from simple to more complex configurations. We considered the following unit configurations: 32, 64, 128, 256, 512, and 1024. When utilizing two hidden layers, the number of units in each layer should remain constant. It is important to note that the other parameters, as described in step (2), have already been determined and should remain unchanged. The primary focus of parameter adjustment in this stage is on the hidden layer structure and the number of units within each hidden layer of the GRU model.

### 3.2. Introducing the Attention Mechanism

The attention mechanism is a widely used technique in deep learning that plays a crucial role in effectively utilizing input information by applying different weights based on specific requirements [38]. By incorporating the attention mechanism into the GRU model, it becomes possible to fully explore the distinctive characteristics of data and use the most significant components within the time series data for modeling purposes. The specific placement of the attention layer can be visualized in Figure 6.

### 3.3. Training Results

After training and adjusting parameters, the GRU + attention model described in Section 3.2 achieved the training losses depicted in Figure 7.

From Figure 7, the early stopping mechanism efficiently terminates the model training process, reducing computation time and mitigating the issue of overfitting while ensuring good training results. The weight coefficient corresponding to the second round is used in the final model.

The model structure consists of a single hidden layer of GRU with a total of 32 hidden units. The difference of the number of training parameters between the GRU model and the GRU + attention model is illustrated Figure 8.

From Figure 8, when the number of units per layer in the GRU model remains constant, the training parameters of the two-hidden layer model are approximately 2.6 to 3.0 times higher compared to the one-hidden layer model. In contrast, the training parameters of the GRU + attention model are almost equivalent to those of the GRU model with a single hidden layer.

### 3.4. Prediction Results

Prediction performance is primarily evaluated based on the *R*^2^ and the degree of correspondence with the hysteretic curve. The average *R*^2^ for the corresponding predicted cases is shown in Table 4. The HDR constitutive model’s structure is defined as a single hidden layer of GRU with 32 units, and the introduction of the attention mechanism between the GRU layer and the output layer. The predicted results at 290.65 K are illustrated in Figure 9.

From Figure 9, the proposed model accurately predicts the stress-strain behavior of each stage of HDR under both amplitude-increasing and amplitude-decreasing loading, effectively capturing the Mullins effect. The model achieves the desired prediction accuracy after only a few training epochs, highlighting the clear advantages of the GRU + attention method in developing the HDR constitutive model.

## 4. Influence of the Loading Process

The above research has shown that the experienced loading process significantly affects the hysteretic behavior during subsequent cyclic loading, based on the two sequential changes of amplitude from small to large and from large to small. In this section, the influence of the experienced loading process on the constitutive relationship of cyclic loading under a constant amplitude is studied. It applies the HDR constitutive model, which is based on deep learning, and contributes to a significant reduction in test costs through research on the HDR constitutive model.

Two amplitudes, small and large, are considered for comparison with subsequent constant amplitude cyclic loading. Each amplitude undergoes one, two, and three loading cycles at a temperature of 290.65 K and a loading rate of 12 mm·s^−1^. The trained GRU + attention model was then used to predict the influence of the experienced loading process on the subsequent loading, as shown in Figure 10 and Figure 11.

From Figure 10, the Mullins effect disappears after the first cycle of the small amplitude loading process. Additionally, when the subsequent loading amplitude is greater than the experienced maximum amplitude, the Mullins effect is further eliminated after the first cycle of the corresponding larger amplitude.

Similarly, from Figure 11, under the loading process of large amplitude, the Mullins effect is eliminated after the first cycle. However, since the subsequent loading amplitudes are less than the experienced maximum amplitude, the Mullins effect is no longer obvious.

After analyzing both Figure 10 and Figure 11, it can be concluded that the Mullins effect primarily manifests itself during the first loading cycle within the corresponding amplitude. The stress-strain relationship remains the same after the first loading cycle.

Similar large amplitude loading processes are likely to occur during a near-fault earthquake due to the obvious influence of the experienced maximum amplitude on the HDR constitutive model. Therefore, when exposed to nearly-fault, large-amplitude seismic pulses, HDR tends to experience initial large-amplitude cyclic loading before transitioning into a state without the Mullins effect. It is important to note that the shear stress generated during the first cyclic loading is greater than that in the subsequent cyclic loading. This should be considered in a seismic design.

## 5. Conclusions

This study employs a deep learning technique to develop an HDR constitutive model that incorporates effects of temperature, rate, experienced maximum amplitude, and the Mullins effect of HDR. The primary findings are summarized as follows:The mechanical performance of HDR is significantly influenced by these factors. The Mullins effect is inevitable when HDR is used for seismic isolation.For test data with distinct time series characteristics, RNN is the preferred method in deep learning. LSTM and GRU have improved capabilities in learning time series data. In this study, the GRU model is used to learn the compression-shear test data of HDR specimens, and the attention mechanism was introduced. As a result, the GRU + attention model is developed, providing a new perspective for research on the HDR constitutive model.The GRU + attention model demonstrates impressive predictive capabilities regarding the impact of the loading process. Additionally, the experienced amplitudes directly affect the stress-strain correlation of HDR during subsequent loading procedures.

## Figures and Tables

**Figure 1 polymers-16-00567-f001:**
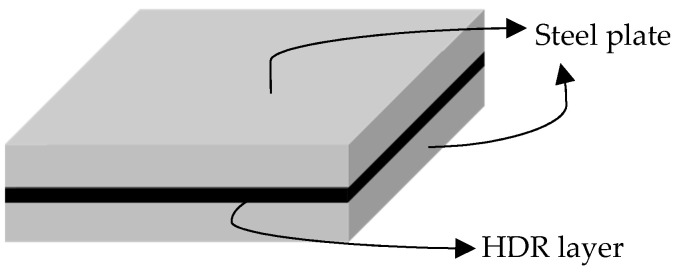
HDR specimen.

**Figure 2 polymers-16-00567-f002:**
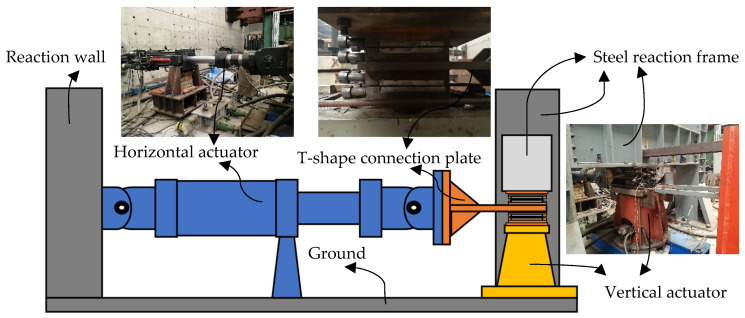
Test loading equipment.

**Figure 3 polymers-16-00567-f003:**
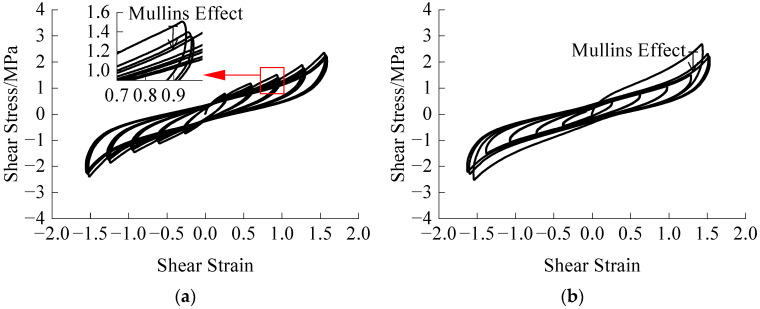
Stress-strain relationships of two loading modes: (**a**) Amplitude-increasing cyclic loading; (**b**) Amplitude-decreasing cyclic loading.

**Figure 4 polymers-16-00567-f004:**
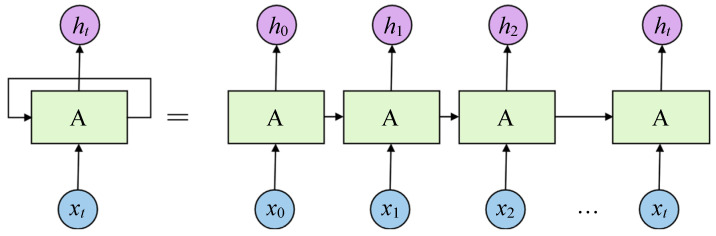
Expanded RNN structure.

**Figure 5 polymers-16-00567-f005:**
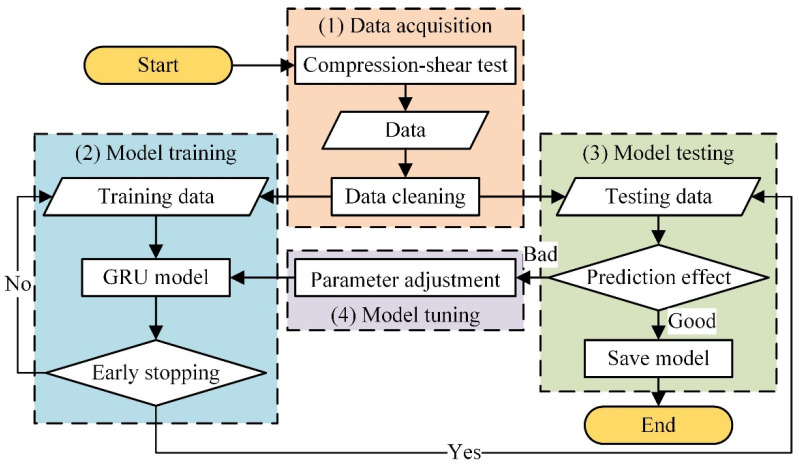
Flow chart of GRU model construction.

**Figure 6 polymers-16-00567-f006:**
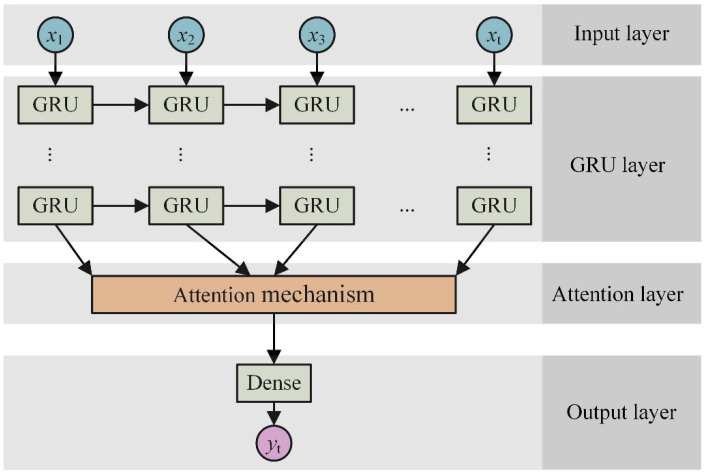
Attention mechanism introducing location.

**Figure 7 polymers-16-00567-f007:**
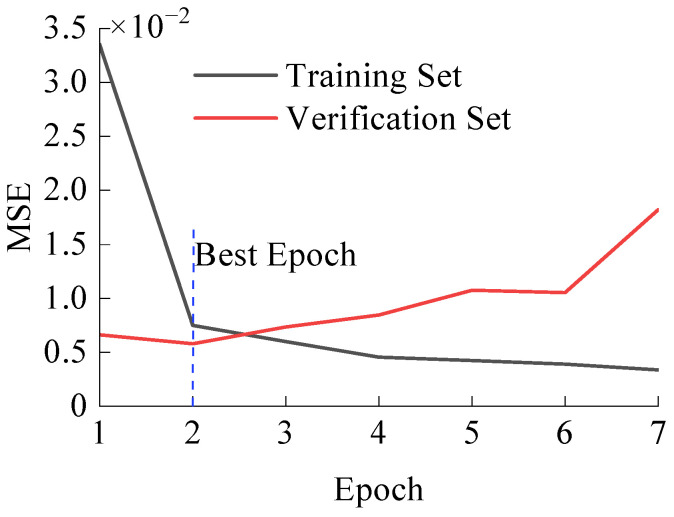
GRU + attention model training losses.

**Figure 8 polymers-16-00567-f008:**
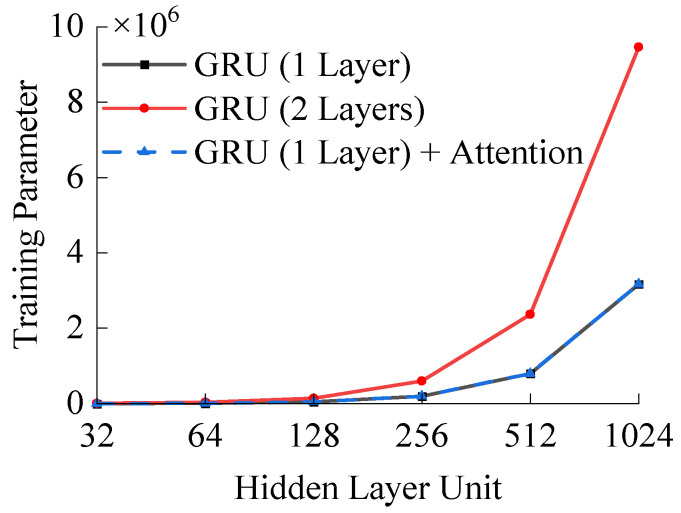
Comparison of the number of training parameters.

**Figure 9 polymers-16-00567-f009:**
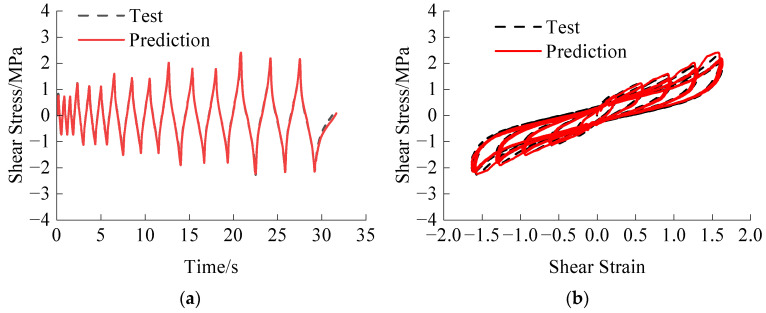
GRU + attention model predictions: (**a**) Shear stress time history of amplitude-increasing loading; (**b**) Stress-strain relationship of amplitude-increasing loading; (**c**) Shear stress time history of amplitude-decreasing loading; (**d**) Stress-strain relationship of amplitude-increasing loading.

**Figure 10 polymers-16-00567-f010:**
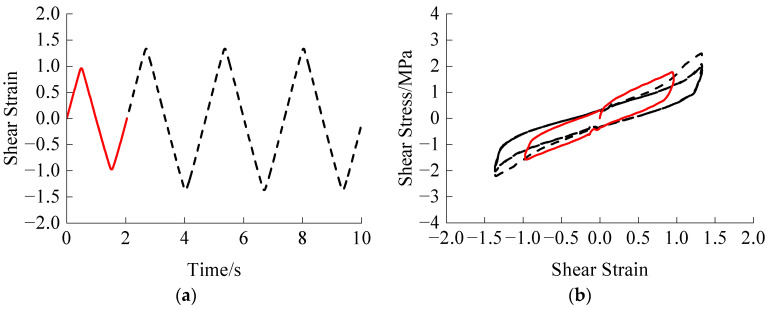
Small amplitude experienced loading: (**a**) Shear strain time history of one experienced loading cycle; (**b**) Stress-strain relationship of one experienced loading cycle; (**c**) Shear strain time history of two experienced loading cycles; (**d**) Stress-strain relationship of two experienced loading cycles; (**e**) Shear strain time history of three experienced loading cycles; (**f**) Stress-strain relationship of three experienced loading cycles. The red lines represent the experienced loading parts, while the black dashed lines represent the subsequent loading parts.

**Figure 11 polymers-16-00567-f011:**
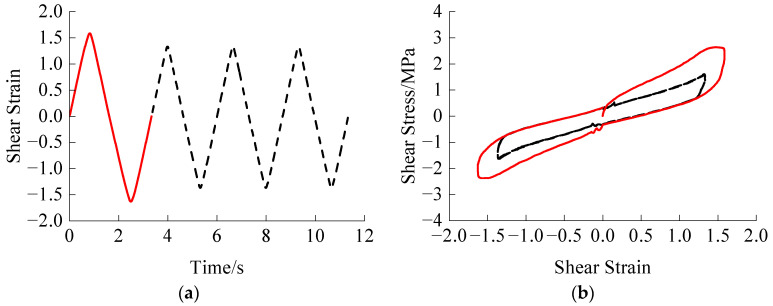
Large amplitude experienced loading: (**a**) Shear strain time history of one experienced loading cycle; (**b**) Stress-strain relationship of one experienced loading cycle; (**c**) Shear strain time history of two experienced loading cycles; (**d**) Stress-strain relationship of two experienced loading cycles; (**e**) Shear strain time history of three experienced loading cycles; (**f**) Stress-strain relationship of three experienced loading cycles. The red lines represent the experienced loading parts, while the black dashed lines represent the subsequent loading parts.

**Table 1 polymers-16-00567-t001:** RNN model training cases.

Parameters	Units	Values				
Temperature	K	253.15	268.15	283.15	298.15	313.15
Loading rate	mm·s^−1^	1	2	4	8	16
Loading amplitude	%	40	80	120	160	200

**Table 2 polymers-16-00567-t002:** RNN model testing cases.

Parameters	Units	Values					
Temperature	K	260.65	275.65		290.65		305.65
Loading rate	mm·s^−1^	12					
Loading amplitude	%	40	80	120		160	200

**Table 3 polymers-16-00567-t003:** GRU parameters.

Parameters	Values
Batch size	64
Initial learning rate	0.001
Verification set percent	20%
Optimizer	Adam
Early stopping epochs	5

**Table 4 polymers-16-00567-t004:** Average *R*^2^ of prediction.

Hidden Units	32	64	128	256	512	1024
Values						
Hidden Layers						
GRU (1 Layer)	0.9927	0.9899	0.9918	0.9956	0.9937	0.9960
GRU (2 Layers)	0.9954	0.9942	0.9949	0.9958	0.9956	0.9939
GRU (1 Layer) + attention	0.9906	0.9918	0.9959	0.9959	0.9965	0.9955

## Data Availability

Data are contained within the article.

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
