# Peer review of "Developing a New Constitutive Model of High Damping Rubber by Combining GRU and Attention Mechanism"

_polymers, 2024, doi:10.3390/polym16050567_

Round 1

Reviewer 1 Report

Comments and Suggestions for Authors

This paper is a good example of the application of AI in developing predictive models for material behavior. 

On the other hand, it suffers from a frequent problem in AI-based work: the great dependence on the initial choices made by authors. I suggest the authors mention at least two kinds of experimental conditions they omitted: i) rubber chemical composition; ii) ambient relative humidity recently shown to produce electrification of rubber under mechanical action.

Comments on the Quality of English Language

The language is adequate.

Reviewer 2 Report

Comments and Suggestions for Authors

1.  The researchers addressed the main research problem, how to develop a new constitutive model of High Damping Rubber (HDR) that accurately predicts its mechanical behavior, taking into account crucial factors such as temperature, rate, experienced maximum amplitude, and the Mullins effect of HDR.

2. The topic is both original and relevant in the field, as it addresses a specific gap in the literature by proposing a deep learning approach that integrates GRU and attention mechanism to construct the HDR constitutive model. This innovative method reduces model complexity and computation time while maintaining good prediction performance, offering a new approach to constructing the HDR constitutive model.

3. The research adds to the subject area by presenting a novel deep learning approach that accurately predicts the mechanical behavior of HDR specimens, taking into account crucial factors such as temperature, rate, experienced maximum amplitude, and the Mullins effect of HDR. This approach reduces model complexity and computation time while maintaining good prediction performance, offering a new approach to constructing the HDR constitutive model.

4. The methodology appears to be well-designed and executed, but the authors could consider further controls to ensure the accuracy and reliability of their results. For example, they could consider using a larger sample size or conducting additional experiments to validate their findings.

5. The conclusions are consistent with the evidence and arguments presented, and they effectively address the main question posed by the research. The authors demonstrate that their proposed deep learning approach accurately predicts the mechanical behavior of HDR specimens, taking into account crucial factors such as temperature, rate, experienced maximum amplitude, and the Mullins effect of HDR.

6. The references are appropriate and provide a comprehensive overview of the relevant literature on the topic.

7. The tables and figures are well-designed and effectively illustrate the key findings of the research. The flow chart of the GRU model construction and the training losses depicted in Figure 7 are particularly helpful in understanding the methodology and results of the study.

Overall, the paper is well-structured, and the methodology is sound. The research makes a valuable contribution to the field by introducing a data-driven approach to HDR constitutive modeling. Addressing minor considerations in methodology transparency and figure captions would further enhance the clarity of the paper.

Reviewer 3 Report

Comments and Suggestions for Authors

In this manuscript, the authors investigate a deep learning based model of high damping rubber. The proposed model employing the gate recurrent unit and attention mechanism showed reduction of model complexity and computation time. The model was then used to analyze the impact of experienced maximum amplitudes and cycles on the high damping rubber processes.

The thorough study of high damping rubber using the combined deep learning approach suggested in this article can make a case for publication. I have a couple of suggestions for the authors:

1.       In section 2 regarding the compression-shear test, it is unclear if this involved experiments or was fully computational. Although it is mentioned that the test was performed by the code, it also includes a part where the test loading equipment are described – please add clarification for this.

2.       In section 4, it is mentioned that previous research has shown the significant impact of loading process. Please provide a relevant reference for this claim.

Overall, the article is well-written with detailed description of model and method, presents data clearly, and is well cited. I would like to recommend this article for publication.

Reviewer 4 Report

Comments and Suggestions for Authors

This article is dedicated to the mechanics-based constitutive model of High Damping Rubber. The article is a complete original research and corresponds to the topic of the journal.

The text of the article does not clearly reflect the purpose of the work and its novelty. 

The quality of representation of illustrative material can also be improved. For example, table 2 needs to be redone because the columns are not readable.
